# Facile Fabrication of Microfluidic Chips for 3D Hydrodynamic Focusing and Wet Spinning of Polymeric Fibers

**DOI:** 10.3390/polym12030633

**Published:** 2020-03-10

**Authors:** Akin Gursoy, Kamran Iranshahi, Kongchang Wei, Alexis Tello, Efe Armagan, Luciano F. Boesel, Fabien Sorin, René M. Rossi, Thijs Defraeye, Claudio Toncelli

**Affiliations:** 1Empa, Swiss Federal Laboratories for Materials Science and Technology, Laboratory for Biomimetic Membranes and Textiles, Lerchenfeldstrasse 5, CH-9014 St.Gallen; Switzerland; akin.gursoy@empa.ch (A.G.); kamran.iranshahi@empa.ch (K.I.); kongchang.wei@empa.ch (K.W.); talexis@student.ethz.ch (A.T.); efe.armagan@empa.ch (E.A.); Luciano.Boesel@empa.ch (L.F.B.); rene.rossi@empa.ch (R.M.R.); Thijs.Defraeye@empa.ch (T.D.); 2Laboratory of Photonic Materials and Fibre Devices (FIMAP), Institute of Materials, Ecole Polytechnique Fédérale de Lausanne (EPFL), 1015 Lausanne, Switzerland; fabien.sorin@epfl.ch

**Keywords:** chip fabrication, fiber wet spinning, hydrodynamic focusing, computational flow modeling, textile industry

## Abstract

Microfluidic wet spinning has gained increasing interest in recent years as an alternative to conventional wet spinning by offering higher control in fiber morphology and a gateway for the development of multi-material fibers. Conventionally, microfluidic chips used to create such fibers are fabricated by soft lithography, a method that requires both time and investment in necessary cleanroom facilities. Recently, additive manufacturing techniques were investigated for rapid and cost-efficient prototyping. However, these microfluidic devices are not yet matching the resolutions and tolerances offered by soft lithography. Herein, we report a facile and rapid method using selected arrays of hypodermic needles as templates within a silicone elastomer matrix. The produced microfluidic spinnerets display co-axially aligned circular channels. By simulation and flow experiments, we prove that these devices can maintain laminar flow conditions and achieve precise 3D hydrodynamic focusing. The devices were tested with a commercial polyurethane formulation to demonstrate that fibers with desired morphologies can be produced by varying the degree of hydrodynamic focusing. Thanks to the adaptability of this concept to different microfluidic spinneret designs—as well as to its transparency, ease of fabrication, and cost-efficient procedure—this device sets the ground for transferring microfluidic wet spinning towards industrial textile settings.

## 1. Introduction

Different from conventional wet spinning where the precipitation of polymer fibers occurs in coagulation baths, microfluidic wet spinning confines polymer and coagulant solutions within co-axial laminar flows. This technology permits a higher control of the diffusion interface amid the two laminar flows, thus allowing the solidification of polymer fibers in a more regulated manner [1]. Several studies have prompted microfluidic wet spinning of fibers featuring different mechanical properties and complex morphologies [2,3]. This was achieved by varying the hydrodynamic shear depending on the core/sheath flow ratio (hydrodynamic focusing) [4] or by opportunely modifying the design of the microfluidic spinneret [5].

Conventional wet spinning of fibers entails a broad range of polymers, namely acrylics, rayon, polyurethane-urea copolymers, aromatic polyamides, and polyacrylonitrile, which have been routinely utilized in the textile industry [6]. Conversely, the transfer of microfluidic wet spinning to the textile industry has been severely hindered by the lack of high-throughput and cost-effective microfluidic spinnerets. Indeed, microfluidic spinnerets for this purpose should be able to establish 3D hydrodynamic flow focusing, thereafter enabling the production of radially uniform fibers [7]. Furthermore, an ideal spinneret should allow facile tuning of the polymer coagulation kinetics by adjusting the channel configurations and flow conditions. This adaptability could avoid the preparation of different devices for producing different fibers, thus ensuring the broad range of applications of a single device. In addition, transparency is another desirable property for such a device, which allows on-line monitoring of the fiber production process.

Despite the existence of many fabrication methods for microfluidic wet spinning devices, none of them can fulfill all the above-mentioned requirements. For instance, soft lithography has been widely used in a laboratory setting for producing microfluidic chips [8,9,10], but involves tedious procedures including mask design and fabrication, plasma bonding, as well as time-consuming slab alignment [11,12]. Alternatively, subtractive manufacturing techniques, such as laser cutting and micro-milling, can shorten the device production procedure [13,14,15,16], but usually require extensive device engineering [14]. Micro-drilling can generate circular channels, but the limited length of the drill bit has restricted the resultant spinnerets for fast coagulating fiber kinetics [13].

Recent advances in additive manufacturing have allowed rapid prototyping of devices with high versatility and precision [17]. While conventional fused deposition modeling (FDM) 3D printers fail to produce microfluidic coaxial reactors due to the inherent x-y low resolution [18,19,20,21], digital light processing (DLP) and masked stereolithography (MSLA) printers have recently shown the possibility to produce transparent microfluidic devices with the single-pixel resolution higher than 47 µm [22,23,24,25,26,27]. However, the channel geometry [27], produced in a layer-by-layer fashion, may affect the flow behavior and therewith the resulting fiber morphology, similar to what was shown by using chevrons [28]. Meanwhile, this effect can be further provoked by the presence of pixelated channel walls [17,23,24]. Although multiphoton lithography solves many of these critical aspects [7] and gives resolutions up to a few hundred nanometers, the equipment cost is too prohibitive for a steady transition to industrial settings.

In this study, we propose a novel fabrication platform for microfluidic wet spinning via the development of a simple and scalable templating method for the production of a spinneret. Templating methods have been used to fabricate devices consisting of tapered glass capillaries molded within transparent silicone-based elastomers [29,30,31,32,33] or thermoplastic blocks produced by micromachinery [13] or stereolithography [34]. Such microfluidic devices have shown smooth inner channel walls and precise control of 3D flow focusing. However, compared to polydimethylsiloxane (PDMS) microfluidic devices, the assembly of glass capillary-based microfluidic devices relies much more on experience and manual operations, thus limiting their commercial applications [35]. Herein, we present a templating method based on PDMS elastomers in combination with an array of disposable stainless-steel hypodermic needles. While preserving the ease of fabrication of other templating methods described above, co-axial alignment of sheath and core channels is easily achieved during PDMS curing. The residence mixing time can be easily adjusted not only by the flow rate but also by positioning the core needle at different distances along the microfluidic device. As an additional advantage, the transparency along the mixing region permits real-time visual monitoring of the fiber production process. Collectively, this method shows great potential in making microfluidic devices for the textile industry and non-microfluidic specialized laboratories. 

To demonstrate the versatility and potential for industrial applications of our platform, we developed a lab-based microfluidic wet spinning stage connected with a tensile drawing apparatus. We picked a thermoplastic polyurethane formulation (BASF’s Ellastolan series) for the spinning trials since they are industrial formulations used in a range of applications from nonwoven fabrics to injection molding. The resultant fiber diameter, surface morphology, and porosity differ by varying the extent of hydrodynamic focusing.

Furthermore, we extended the versatility of this device with the spinning of poly(acrylonitrile) (PAN) and poly(vinyl alcohol) (PVA) fibers, [3,7]. We have shown that, with the same device, micrometric fibers with diameter down to single digit can be fabricated at high hydrodynamic focusing ratios.

## 2. Materials and Methods 

### 2.1. Microfluidic Spinneret Fabrication

The design and fabrication of the spinneret are shown in Figure 1. Two different types of devices were built to demonstrate the versatility of the fabrication method. The first reactor (utilized for the microfluidic wet spinning experiments) displays one core flow channel and a single set of sheath flow channels. The second reactor type displays two different sets of sheath flow channels and one core flow channel, resulting in a multicore flow channel configuration (Appendix A).

As depicted in Figure 1, the fabrication consists of three phases: preparation, molding, and assembly. Initially, 21G (Sterican^®^ BBraun, Carl Roth, Arlesheim, Switzerland) and 25G (Sterican^®^ BBraun, Carl Roth, Arlesheim, Switzerland) hypodermic needles were blunted and cut to length by a rotary tool (Proxxon MICROMOT 60/E, Föhren, Germany) attached with a cutting disc (Aluminium Oxide, Proxxon, Wecker, Luxembourg). These needles were then cleaned from burs if necessary, with a cutting disc, cleaned with water and acetone (Sigma-Aldrich, ≥99.9%, HPLC grade, Buchs, St. Gallen, Switzerland) and left to dry. 

During the molding phase, two-component Dow Corning Sylgard 184 (Suter-kunststoffe AG, Fraubrunnen, Switzerland) was used thoroughly with a 1:10 cross-linker/prepolymer ratio. The resin was transferred to 15 mL Falcon tubes (Falcon, Faust, Schaffhausen, Switzerland) and centrifuged for 30 s to remove air bubbles trapped inside. A glass Petri dish with the appropriate size was used as a negative mold and filled with the resin until reaching half of the desired spinneret thickness. The resin was partly cured for 5 min at 150 °C in an oven (FD Classic, Binder, Tuttlingen, Germany). 

After the first PDMS layer was partly cured, a 25G needle was slightly inserted into a 21G needle and laid in the middle of the dish. Two 21 G needles were positioned perpendicularly to the axially aligned 25G and 21G needles to a cross shape. Afterward, the rest of Sylgard 184 resins was poured in the Petri dish. The air bubbles inside the needles were removed by placing the glass dish into a desiccator connected to a vacuum pump. Full curing of the first and the second resin layers was attained by placing it in the oven and heating up to 150 °C for 10–15 min. 

After cooling down to room temperature (RT), the PDMS with needles embedded inside was removed from the glass dish by using methanol (Sigma-Aldrich, ≥99.9%, HPLC grade, Buchs, St. Gallen, Switzerland) as a lubricant and a spatula for detaching it from the glass (assembly phase). With methanol as a lubricant, PDMS was cut by a utility knife to expose the needles, which were subsequently removed with tweezers, yielding the microfluidic spinneret.

### 2.2. Computational Model and Simulation

A two-dimensional, axisymmetric model is implemented in COMSOL Multiphysics (version 5.4a) to investigate the laminar flow behavior and the resultant hydrodynamic focusing in the microfluidic reactor. Finite element modeling is used to solve the governing equations. An axisymmetric model is justified by the symmetry of the geometry. A schematic of the computational model together with the applied boundary conditions are provided in the Appendix A. The computational model includes two cylindrical tubes representing the core needle and reactor tube (Appendix A). The geometrical dimensions of the model have been set the same as in the experimental setup. The outlet of the tube is taken sufficiently far away from the mixing section to avoid any influence of the boundary on the hydrodynamic focusing. The location where the hydrodynamic focusing was evaluated is similar in experiments and simulations, specifically at 10 mm after core needle tip. The fluid is modeled with the same fluid properties of PEG (Mw 400 Da) for both core and sheath flows. The dynamic viscosity, density, and diffusion coefficient are 0.115 Pa⸱s (at 20 °C), 1260 kg/m^3^, and 7 × 10^−10^ m^2^/s, respectively.

The computational domain includes two separate flow inlets for core and sheath flow (Appendix A). Two different fluid speeds are imposed on the inlets based on sheath and core flow rates. In order to demarcate the core–sheath flow boundary, since the same fluid is used, a diluted species marker is used on the inlets with the concentration of 0.1 mol/m^3^ (C% = 100%) and 0 mol/m^3^ for core and sheath inlets, respectively. Atmospheric pressure condition is imposed at the outlet boundary. Appropriate grids were built based on a grid sensitivity analysis. Grid refinement has been performed toward the core needle, to enhance numerical accuracy and stability, and toward the measuring section to have a better resolution in the results. Finally, laminar flow and transport of diluted species were solved together as a stationary problem. Different combinations of core and sheath flow rates were evaluated. The tolerances for convergence were selected based on sensitivity analysis in such a way that increasing the tolerance further did not change the results (core flow diameter) anymore. The simulation model (Figure 2) was validated on the basis of the experimental measurements (Figure 3E). More details about the simulation together with the validation for a higher total flow rate are available in the Appendix A.

### 2.3. Flow Experiment

Laminar flow in the reactor, as well as hydrodynamic focusing, has been studied by flow experiments and visualized with an overhead microscope (Leica DMS300 with LED3000RL diffusor, Wetzlar, Germany) (Figure 1). For this experiment, water and PEG (*M*_w_ 400 Da, Sigma-Aldrich, bioultra, Buchs, St. Gallen, Switzerland) were used as the model fluids. A few drops of saturated aqueous methylene blue (Sigma-Aldrich, Buchs, St. Gallen, Switzerland) solution was utilized to enhance the boundary region amid the core and sheath flow. The two fluids were loaded into syringes (Injekt^®^ BBraun, Carl Roth, Arlesheim, Switzerland) that were attached to three microfluidic syringe pumps (neMESYS 290N powered by BASE 120, Cetoni, Korbussen, Germany). The three syringes linked to the core flow channel and the two sheath flow channels were connected to 1 mm inner diameter silicone tubes by utilizing 18G (Sterican^®^ BBraun, VWR) blunt needles. The microfluidic spinneret was prepared by firstly tapering 21G and 25G needles with a rotary tool and then inserting them into the respective channels. All inlets were connected to the pumps via silicone tubes. The study was performed during continuous flow and microscopy images were taken for analysis and the diameter of the core flow was measured using imaging software (Leica Application Suite LAS EZ v3.4.0) at three different locations along the main channel. For this study, the total flow rate of 300 µL/min was chosen with a sheath/core flow ratio (r) of 0.4, 0.5, 1, 2, 5, 10, 20, 50, 100. Four additional high-speed tests were performed at 1000 µL/min with sheath/core flow ratio (r) of 10, 50, 100, 200.

### 2.4. Thermoplastic Urethane (TPU) Fiber Spinning

Syringes and tubes in this section were connected similarly to the flow experiments. The spinning dope consisted of 15% *w/v* of TPU (Ellastolan C 78A15, BASF, Ludwigshafen, Germany) in DMF (Sigma-Aldrich, >99.5%, ACS Reagent, Buchs, St. Gallen, Switzerland). To prepare this solution, TPU was dissolved in DMF by heating and stirring until reaching a clear solution, which was then cooled down to RT to yield a slightly cloudy viscous solution. Meanwhile, 6% *w/v* PEO (900kDa, Sigma-Aldrich, Buchs, St. Gallen, Switzerland) in distilled water was prepared by stirring at 70 °C until complete dissolution. The solution was then diluted 1:1 with DMF to obtain a cloudy solution of 3% *w/v* PEO, which was subsequently used for sheath flow. 

Sheath fluid was filled into two 20 mL disposable syringes (Injekt^®^ BBraun, Carl Roth, Arlesheim, Switzerland) connected to the sheath flow channels of the reactor by a 21G needle while spinning dope was filled into a 5 mL disposable syringe (Injekt^®^ BBraun, Carl Roth, Arlesheim, Switzerland) connected to the core flow channel with a tapered 25G needle. 

The nozzle was inserted into the PDMS chip until it is 5 mm away from the T-junction (Figure 1D, inset) and the chip was placed at the reactor holder of the mini-wet-spinner and lowered until the outlet channel was submerged in the water bath. Sheath flow was set at 200 µL/min until the air is removed from the spinneret and lowered to 20 µL/min. Core flow was started at a flow rate within the range of 100–200 µL/min and maintained until the fluid reached the needle. At this point, the pumps were set to a total flow rate of 500 µL/min, the collector motors were started and the spinning process began. A total of four different sheath/core flow ratios (r) were used, namely 2, 20, 50, and 100. For r = 100, an additional experiment was performed with a total flow rate of 1000 µL/min.

A mini-wet-spinner was used to collect and slightly draw the spun fiber with the microfluidic spinneret directly mounted on it (Figure 4A). The three motors of the mini-wet-spinner can be controlled directly by a Raspberry Pi^®^ powered board and touch screen user interface. Initially, the rotation speeds of these motors were set by converting the flow rate into mm/min and were configured using in-house developed software. During the fabrication, the touchscreen was used to monitor parameters such as speed, the traverse of the winding spool, temperature and flow of the drying fan, and total fiber collected at each spool, as well as to put in online commands to fine-tune speeds. 

The initial motor speeds were set 1.1 times of the flow speed in mm/min, calculated from the supplied flow rate. The second motor was adjusted to 1.1 times the speed of the first motor to create tension and to collect the fiber without a sag between two motors. During the fine-tuning of the first motor speed, the rotation of the second motor is automatically calculated and adjusted by the software to keep the pre-determined drawing ratio.

### 2.5. Versatility Testing

6% *w/v* PVA (Mowiol 40-88, Sigma-Aldrich) in water and 5% *w*/*v* PAN (100kDa, Sigma-Aldrich, Buchs, St. Gallen, Switzerland) in DMF were used for core flows, while PEG (*M*_w_ 3350 Da) in ethanol (EtOH, Sigma-Aldrich, >99.8%, ACS Reagent, Buchs, St. Gallen, Switzerland) at 20% *w/v* or in EtOH/DMF (2:1) at 13% *w*/*v* were used for sheath flow to induce the coagulation of PVA and PAN, respectively. The microfluidic device used in these experiments only differed in the main channel diameter of 1.2 mm, which was derived from the outer diameter of 18G needles (Sterican^®^ BBraun, Carl Roth, Arlesheim, Switzerland). 21G and 25G hypodermic needles were used as sheath and core nozzles respectively in a spinneret with the same geometry as previously described. During the experiment, the total flow rate was set as 500 µL/min for both fibers. PVA fibers were spun into an acetone (Fischer Chemicals, HPLC grade, Reinach, Switzerland) bath and collected manually with a tweezer. PAN fibers were spun at r = 20 and 100, and collected by the mini-wet-spinner.

### 2.6. SEM Analysis of Produced Fibers

The SEM images of the produced fibers were taken with a JEOL Hitachi S-4800 (Chiyoda, Tokio, Japan) SEM at 2 keV with 5 nm of Au-Pd sputtering prior to loading in SEM sample chamber. Cross-sections were obtained by cutting the fibers with scissors in a liquid nitrogen bath and imagined by tilting the SEM stage.

## 3. Results

### 3.1. PDMS Chip Fabrication

The microfluidic spinnerets we are demonstrating in this study were manufactured by using hypodermic needles as templates of the channels. These needles can be sourced from commercial suppliers and are offered with dimensional tolerances as low as ±12.7 µm (34G), making them excellent templating materials for smooth and circular microfluidic channels. The needles can be cut into required lengths with a rotary tool to obtain the desired spinneret channel templates and tapered to avoid sharp edges that would lead to flow instabilities. Co-axial alignment of these needles is achieved by selecting commercial needles with different Gauges (G) and coupling co-axially the fitting ones. Specifically, two pairs of needles were used for templating the co-axial channels, i.e., 25G/21G pair and 21G/18G pair (Figure 1A). 

Sylgard 184 was used as a thermoset resin for producing PDMS chips, since it can be molded for fast production of transparent objects, with a curing time of 10 min at 150 °C. The needles were employed as negative molds, encapsulated between two layers of PDMS for the construction of the microfluidic spinneret (Figure 1). This avoids the time-consuming steps commonly involved in conventional soft lithography, with consequently reduced fabrication time down to only 30 min. Moreover, thanks to the elastic nature of the PDMS matrix, the assembled spinnerets can ensure leakage-free sealing without using microfluidic fittings or interconnects.

During the initial trials, several spinnerets with different designs have been fabricated, including those with thicker channels or multicore spinnerets (see Appendix A for details, Appendix A). However, further tests were carried out with a cross-shaped spinneret using 21G and 25G needles as templates (Figure 1, Preparation). Core and sheath templating needles were deposited on cross-linked Sylgard and subsequently embedded by in-situ cross-linking of the same material (Figure 1, Molding). After removing the templating needles, 25 G needle was placed as the core channel and two 21G needles were placed as sheath channels (Figure 1 Assembly). The resultant spinneret has a nozzle diameter of 280 µm, side inlet diameter of 510 µm and main channel diameter of 800 µm. Leakage occurred only in the case of pressure overshoot caused by unexpected channel clogging. 

### 3.2. Fluid-Dynamics Simulation

Fluid flow characteristics were analyzed by using laminar fluid dynamics simulations with the finite element method (FEM). This computational model aims to evaluate the impact of different sheath/core flow rate ratios (from here on referred as “r”, r = Q_sh_/Q_c_, where Q_sh_ and Q_c_ represent the volume flow rate of the sheath and core flows, respectively) on co-axial flows of polymer core and coagulant sheath solution or so-called hydrodynamic focusing. These insights open up opportunities to optimize geometrical parameters of the setup (e.g., core/sheath needles relative length and/or diameter), to define the set of boundary conditions where the microfluidic device can operate to produce fiber in a continuous fashion. Moreover, overall flow stability as well as uniformity around the nozzle, which is critical for 3D hydrodynamic focusing and stable fiber production, were evaluated.

The velocity profiles show the direction and magnitude of the core and sheath flow in different locations (Figure 2A). Mixing flow development and focusing evolution are obvious in these velocity profiles. Simulations have shown that stable focusing can be achieved until r = 100, at which point backflow started to develop with r > 100 (Appendix A). This study was further complemented by 3D modeling of the flow behavior, which showed a stable 3D focusing on the core flow (Figure 2B). To extract the core flow diameter from flow simulations, the concentration of the core flow fluids is used to demarcate the core–sheath flow boundary (Figure 2C). The core–sheath flow boundary is defined for a specific threshold of the initial concentration of the core flow (C%). To this end, three different threshold concentrations (1%, 0.1%, and 0.01% of initial core-flow concentration) were considered to distinguish the core flow boundary. These values were considered based on the experimental measurement uncertainties.

### 3.3. Flow Experiment

The flow experiments employ the constructed four-channel spinneret operated under a microscope. The flow characteristics were visualized by mixing methylene blue and poly(ethylene glycol) (PEG, *M*_w_ 400 Da) in the solution for core flow, while colorless PEG (*M*_w_ 400 Da) was used for sheath flow. PEG was chosen as a model fluid for the flow experiments as its viscosity resembles the one utilized in the subsequent experiments for fiber production. Compared to using water as the model fluid, this enabled us to evaluate the performance of the microfluidic spinneret under the conditions that are more relevant to fiber production. To compare experimental results with simulations, the same total flow rate of 300 μL/min was chosen. Hydrodynamic flow-focusing was characterized by varied sheath/core flow ratio (r) ranging from 0.5 to 200 (Figure 3). Stable core/sheath laminar flows under these flow conditions were observed and representative flow behaviors were imaged (Figure 3A–C). The defocusing of the core flow was observed at r = 0.5 (Figure 3A). With increasing r (r > 0.5), a progressive decrease in core flow diameter was observed and can be nicely tuned from 600 to 50 μm as demonstrated by both simulation and experiments (Figure 3E).

Albeit not stable, the laminar flow could be obtained with such a spinneret at r = 100 with 300 µL/min total flow rate. In addition, stable laminar flow can be maintained at r = 100 when the total flow was increased to 1000 µL/min with the sheath flow being gradually increased up to the targeted value. The spinneret was further pushed to its limit at r = 200 by applying a gradual increase of hydrodynamic focusing and therewith a stable core/sheath laminar flow was achieved (Figure 3D). The results obtained from flow simulations show that the simulated estimation of core flow diameter boundaries is in good agreement with experimental data (Figure 3E). Nevertheless, the lower the r, the closer the experimental value is to the 0% concentration boundary, which is due to the lower error in measuring the boundary experimentally. This implies that the current computational model is reliable and can be used for further evaluations such as geometrical modifications.

### 3.4. Microfluidic Wet Spinning of Thermoplastic Polyurethane Fibers

As a proof of concept, thermoplastic polyurethane fibers were produced by microfluidic wet spinning. 15% *w/v* TPU solution in DMF was used for core flow. 3% *w/v* PEO (*M*_w_. 900 kDa) solution in DMF/H_2_O (1:1) was used as sheath flow. This mixture of solvent and non-solvent in sheath flow ensures slow coagulation of TPU polymers in the core flow [36], therefore preventing nozzle clogging that occurs frequently under fast coagulation processes. Meanwhile, the slow coagulation could reduce porosity [37], leading to the improved mechanical strength of the resultant fibers [4]. Additionally, the possibility to modulate the fiber porosity can be beneficial in tuning the pharmacokinetics for localized drug delivery [38] or in enhancing sensor sensitivity [39] when the resultant fibers are loaded with active drugs or sensing elements, respectively. The addition of high molecular weight PEG in the sheath flow increases the viscosity, lowers the Reynolds number, and increases the pressure applied to the core flow for increased focusing.

To facilitate the fiber spinning and collection, the microfluidic spinneret was coupled to a customized automatic fiber drawing system (called mini-wet-spinner, Figure 4A and Appendix A) equipped with a set of three cylindrical rolls, which can pull the fibers that are being spun into the coagulation bath (i.e., water in this case) from the microfluidic spinneret. This enables continuous spinning and collection of fibers until the complete consumption of the precursor solution in the syringe reservoir, leading to the production of long fibers up to several meters at speeds up to 3.5 m/min. Utilizing this setup, TPU fibers with different diameters were successfully produced with the microfluidic spinneret (Figure 4B–E). As expected, with the highest sheath/core flow ratio, produced fibers have the smallest diameter (Figure 4D). Interestingly, scanning electron microscopy (SEM) images show strikingly different surface patterns for the fibers produced at r = 2 (Figure 4B). Compared to other fibers with a smooth surface, these show wrinkled surfaces with deep grooves roughly oriented along the fiber axis. In addition, the average diameter of these fibers (154 µm, Figure 4B) was close to that of the ones produced at r = 20 (131 µm, Figure 4C). Only a diameter reduction of 15% was induced by the higher r. According to the flow experiments, the diameter of the core flow is 386 μm at r = 2. A reduction (56%) of the core flow diameter to 169 μm was observed at r = 20 (Figure 3). Therefore, higher differences in diameter between fibers spun at r = 2 and r = 20 were expected according to the flow experiments. Their relatively similar size observed by SEM (Figure 4B,C) may suggest that the fibers spun at r = 2 may collapse due to the highly porous nature. 

To confirm this hypothesis, we performed a cross-section analysis of the fibers. SEM images clearly show a decrease in porosity as the sheath/core flow ratio increases with the exception of r = 2 (Figure 5). Compared to other fibers, these fibers (Figure 5A) show low porosities, but more rectangular fiber cross-section, indicating the collapse of the fibers under observation. This collapse may be due to the fast development of the fiber shell at the core/sheath boundary, followed by shrinkage from the core towards the shell during the spinning procedure.

### 3.5. Versatility in Microfluidic Wet-Spinning of Other Polymer Fibers

The versatility of the microfluidic spinneret was tested by spinning PVA and PAN fibers, which have been previously described in the literature [3,7]. As shown in Figure 6, PVA and PAN fibers were successfully produced with the same device. It is noteworthy that PVA fibers as small as 4 μm in diameter can be produced at r = 100 with a total flow rate of 500 μL/min. Interestingly, this feature matches with previous achievements with more complicated and challenging nozzle designs [7]. The production of different polymeric fibers displaying various coagulation regimes further confirms the versatility and universality of this reactor for its translation to industrial spinning. 

## 4. Conclusions

Herein, we describe a customizable microfluidic spinneret for the scalable production of various polymer fibers. We demonstrated the capabilities of such a device in establishing 3D hydrodynamic flow focusing by using fluid dynamics simulations as well as flow experiments. Moreover, we validate the device for polymer fiber production by spinning a thermoplastic urethane fiber under different hydrodynamic focusing conditions. The device is versatile and other polymers, namely poly(vinyl alcohol) and poly(acrylonitrile) were also spun. We have also shown that the fabrication of individual few-micron-thick fibers can be considered realistic with this device. In addition, the present microfluidic device can be upgraded with two sets of sheath flows and a multicore channel with three differential channel diameters along the main channel. The upgraded version would enable the microfluidic production of complex fibers, including hollow fibers, core–shell fibers and liquid-core fibers. Such a cost, time and labor efficient fabrication of versatile microfluidic devices with these features will ease the translation of microfluidic wet spinning within industrial textile research facilities. Indeed, the fabrication technique developed here can make microfluidic research more accessible for not only the academic field, but also to the industrial sector.

## Figures and Tables

**Figure 1 polymers-12-00633-f001:**
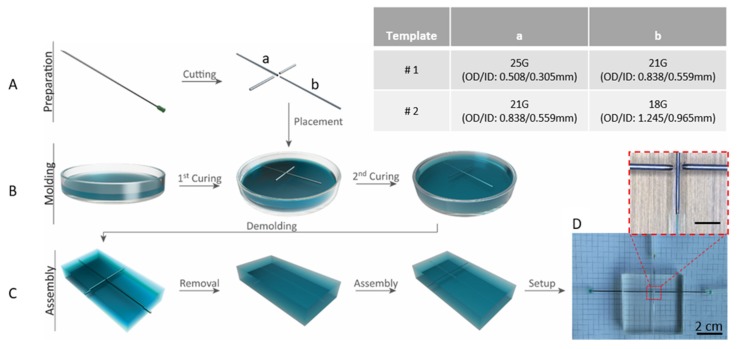
Schematic illustration of the fabrication procedure of the microfluidic spinnerets. (**A**) Preparation of the channel templates with different needles (OD: outer diameter; ID: inner diameter). (**B**) PDMS cross-linking and molding into the device features. (**C**) Assembly of the microfluidic spinneret. (**D**) A representative photograph of the final product with inset showing the detail of the microfluidic channels (inset scale bar: 5 mm).

**Figure 2 polymers-12-00633-f002:**
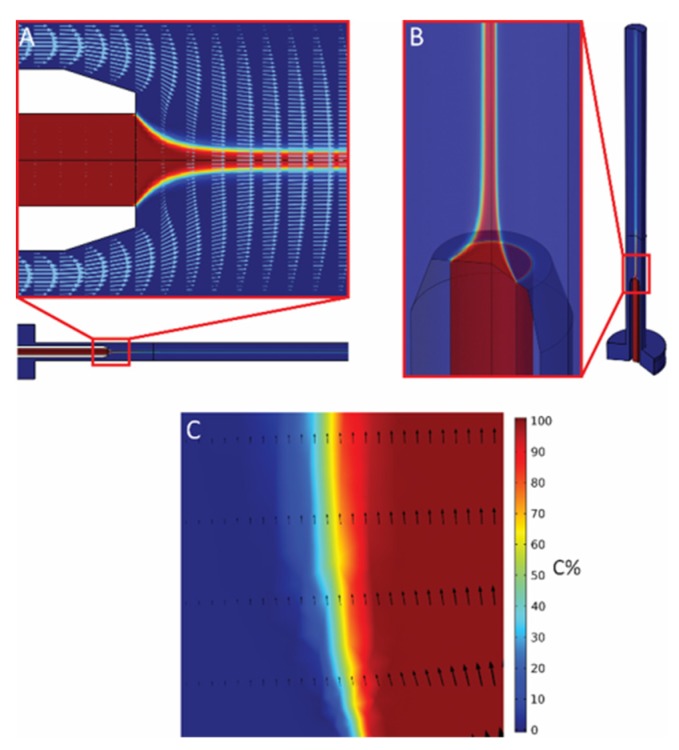
Simulation of the focusing behavior at r = 0.2 with total flow rate of 300 μL/min. (**A**) 2D vector and concentration graph. (**B**) 3D simulation of visualizing flow behavior around the nozzle. (**C**) Concentration gradient at the boundary of the core flow.

**Figure 3 polymers-12-00633-f003:**
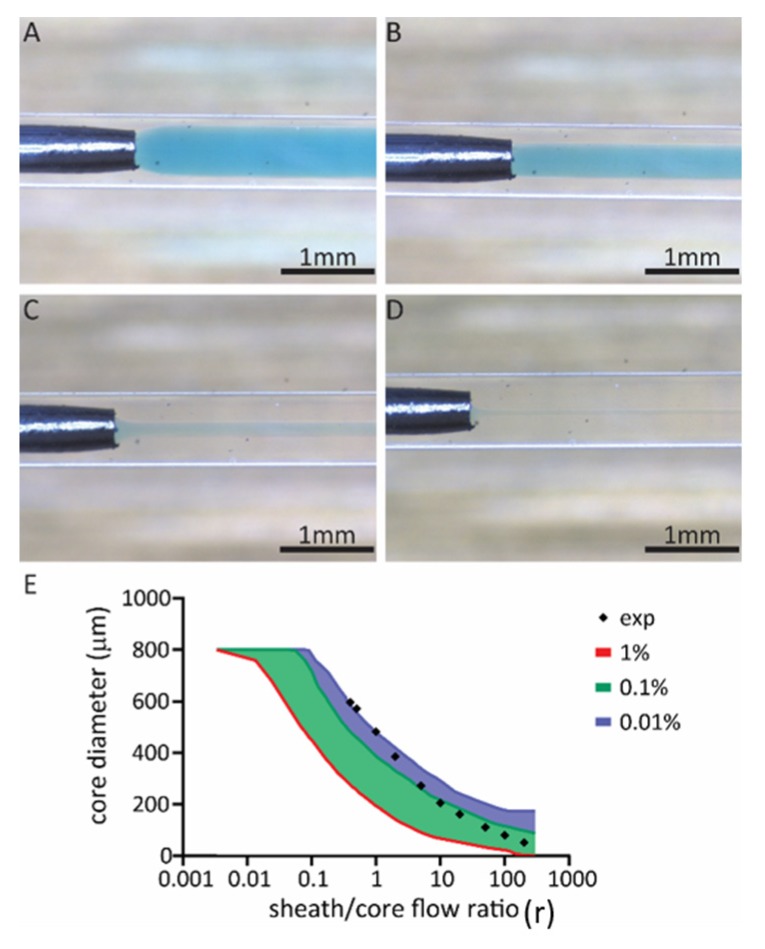
Core/sheath laminar flows in the microfluidic spinneret with constant total flow rate of 300 µL/min and varied sheath/core flow ratio of (**A**) r = 0.5; (**B**) r = 2; (**C**) r = 20. (**D**) Core/sheath laminar flow with a total flow rate of 1000 μL/min (r = 200). (**E**) Comparison of experimental values with simulated boundaries.

**Figure 4 polymers-12-00633-f004:**
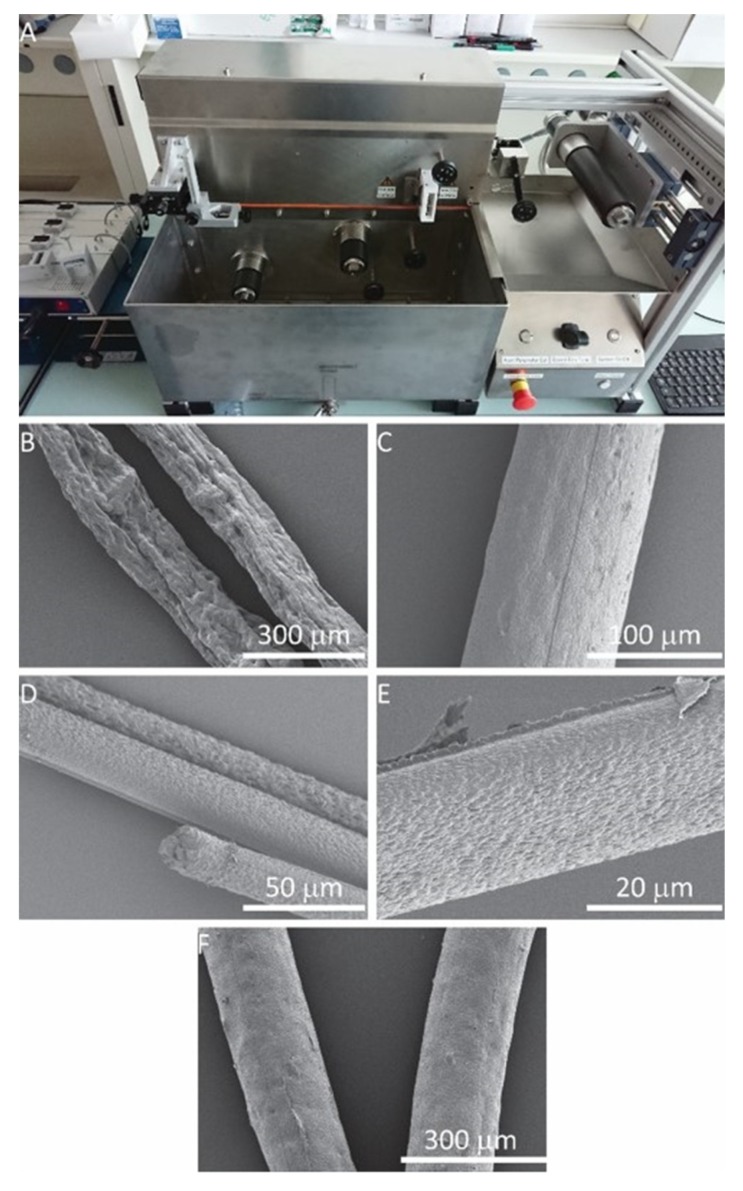
(**A**) Customized mini-wet-spinner used to collect fibers and microfluidic pumps used for flow control (left). (**B**–**E**) SEM images of TPU fibers. (**B**) TPU fiber spun at r = 2. (**C**) TPU fiber spun at r = 20. (**D**) TPU fiber spun at r = 50. (**E**) TPU fiber spun at r = 100. (**F**) TPU Fiber spun at r = 20 using a microfluidic spinneret with the main channel diameter of 1.2 mm.

**Figure 5 polymers-12-00633-f005:**
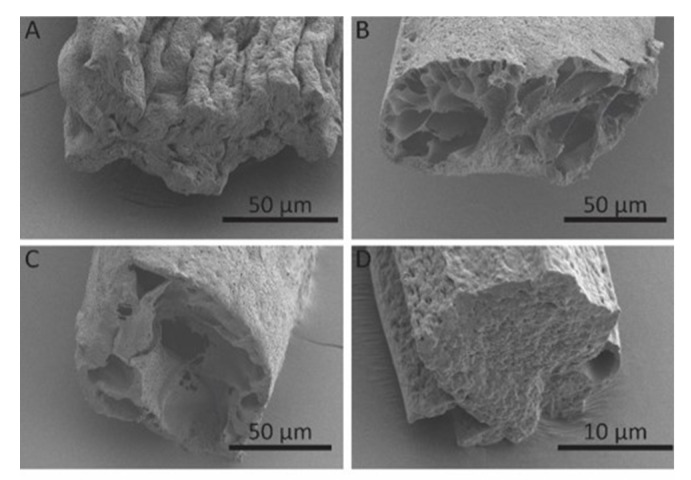
SEM images (cross-sections) of TPU fibers spun at (**A**) r = 2; (**B**) r = 20; (**C**) r = 50; (**D**) r = 100.

**Figure 6 polymers-12-00633-f006:**
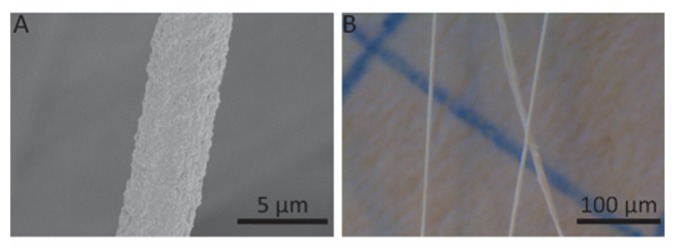
SEM images of other fibers spun with the microfluidic spinneret. (**A**) PVA fiber. (**B**) PAN fiber.

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
