# Peer review of "Facile Fabrication of Microfluidic Chips for 3D Hydrodynamic Focusing and Wet Spinning of Polymeric Fibers"

_polymers, 2020, doi:10.3390/polym12030633_

Round 1
Reviewer 1 Report
This experimental paper reports on the "Facile fabrication of microfluidic chips for 3D hydrodynamic focusing and wet spinning of polymeric fibers". The paper is well-written, well-organised and well-presented. There are few changes required. I think a couple of formatting/technical issues need to be addressed (e.g.; page 4, line 164; reference error). Once dealt with, the manuscript should be ready for publication.
Author Response
The authors thank the reviewer for reviewing the paper and for spotting the formatting issue(s).
These has been corrected in the revised version of the manuscript.
Reviewer 2 Report
This manuscript demonstrates a practical method using selected arrays of hypodermic needles as templates for making PDMS microfluidic spinnert for spinning microfibers. The results from their experiments and computer simulations have seemed good supports for their above idea. However, the following details need be further clear.
(1) During the co-axial alignment of sheath and core channels when PDMS curing, how to avoid the possible gap between the “sheath” needle and “core” needle, therefore to ensure the communication of the sheath flow path and core flow path.
(2) Line 123 on page 3, ‘The resin was transferred to 15 ml Falcon 123 tubes (Falcon, Faust) and centrifuged for 30 seconds to remove air bubbles trapped inside. ’ Could this operation effect?
(3) Line164 on page 4, what’s meaning of that: “measurements (Error! Reference 164 source not found.E). ”?
line 32 on page1,“fiber wet spinning” is repeated.
Author Response
The authors thank the reviewer for the evaluation of this manuscript.
Here below are the point-by-point reply to his/her questions:
(1) During the co-axial alignment of sheath and core channels when PDMS curing, how to avoid the possible gap between the “sheath” needle and “core” needle, therefore to ensure the communication of the sheath flow path and core flow path.
Answer. The possible gap between the "sheath" needles and "core" needle during the PDMS templating procedure, which might be partly occluded by the PDMS elastomer, can be easily removed during the insertion of the sheath flow needle prior to the spinning.
(2) Line 123 on page 3, ‘The resin was transferred to 15 ml Falcon 123 tubes (Falcon, Faust) and centrifuged for 30 seconds to remove air bubbles trapped inside. ’ Could this operation effect?
Answer. Since air bubbles can be easily visible by optical microscopy thanks to the transparency of PDMS, we could check that centrifuging for 30 second could effectively eliminate the air bubbles present within the matrix.
(3) Line164 on page 4, what’s meaning of that: “measurements (Error! Reference 164 source not found.E). ”?
Answer. The authors thanks the reviewer for spotting the issue. It did not appear in the original word manuscript, but it probably occurred during the conversion to pdf.
(4) line 32 on page1,“fiber wet spinning” is repeated
Answer. Amended.